# UNSUPERVISED ANOMALY DETECTION FROM SEMANTIC SIMILARITY SCORES

## ABSTRACT

In this paper we present *SemSAD*, a simple and generic framework for detecting examples that lie out-of-distribution (OOD) for a given training set. The approach is based on learning a semantic similarity measure to find for a given test example the semantically closest example in the training set and then using a discriminator to classify whether the two examples show sufficient semantic dissimilarity such that the test example can be rejected as OOD. We are able to outperform previous approaches for anomaly, novelty, or out-of-distribution detection in the visual domain by a large margin. In particular we obtain AUROC values close to one for the challenging task of detecting examples from CIFAR-10 as out-of-distribution given CIFAR-100 as in-distribution, without making use of label information.

## 1 INTRODUCTION

Anomaly detection or novelty detection aims at identifying patterns in data that are significantly different to what is expected. This problem is inherently a binary classification problem that classifies examples either as in-distribution or out-of-distribution, given a sufficiently large sample from the in-distribution (training set). A natural approach to OOD detection is to learn a density model from the training data and compute the likelihood ratio of OOD examples. However, in practice this approach frequently fails for high-dimensional data (Nalisnick et al. (2019)), where it has been shown that deep generative models can assign higher likelihood to OOD examples than to in-distribution examples. This surprising result is likely the consequence of how existing deep generative models generalise. For example, Variational Autoencoders (Kingma & Welling (2014)) generalise by superperposition of examples, which is a consequence of the stochastic nature of the posterior that can map different examples to the same point in latent space. As superposition is an averaging process that reduces the information content it can be expected that examples of lower complexity than the training examples can map to high likelihood regions in latent space. Note that it is possible for a datapoint to have high likelihood under a distribution yet be nearly impossible to be sampled, a property known as asymptotic equipartition property in information theory Cover & Thomas (2001). For autoregressive generative models, such as PixelCNN (van den Oord et al. (2016)), it has been shown that the pixel-by-pixel generation process is strongly determined by the local surrounding of pixels (Chen et al. (2018)), where the fact that nearby pixels of training examples frequently share the same color can explain why mono-chromatic images are assigned a high likelihood (Nalisnick et al. (2019)). Local pixel correlations also seem to be responsible for the failure of generative models based on Normalising Flows to assign correct likelihood values to OOD examples Schirrmeister et al. (2020).

As a consequence, most of the current OOD detection approaches make use of a score function $s(x)$ to classify test examples as in-distribution or OOD. In case that the examples of the training set are labelled, a simple score can be given by $s(x) = \max_y p(y|x)$, with $p(y|x)$ the softmax probability for predicting class labels, $y \in \{1,..,K\}$ (Hendrycks & Gimpel (2017)). If $s(x)$ is below a threshold the test example is classified as OOD. Labelled data allows to learn representations that are associated with the semantic information shared by the examples in the training set, which can be used for OOD detection. However, the approach suffers from the problem that the scores for in-distribution examples can be widely distributed across the interval of possible score values, $s(x) \in [1/K, 1]$, especially if the number of labels are low and the classification task is hard, which strongly increases the false-positive rate. Consequently, better performance was found for approaches that use labeled data for learning a higher dimensional representation that encodes for

semantic information (Lee et al. (2018b)). In this representation space the in-distribution occupies just a small volume and a random feature vector would be most likely classified as OOD. Another simplification arises if the OOD detection problem is supervised, with some OOD examples labelled as such and contribute to the training set. In this case the OOD detection problem boils down to an unbalanced classification problem (Chalapathy & Chawla (2019)). In general OOD detection benefits from separating the factors of variation for the in-distribution in either relevant (e.g. object identity) or irrelevant (e.g. compression artefacts) using prior knowledge, where the relevant factors are typically those that carry salient semantic information. In line with the arguments put forward by Ahmed & Courville (2020), this separation helps an OOD model to systematically generalise, e.g. whether we are allowed to re-colour or add noise to images for data augmentation. Generalisation over the training set is necessary, as learning under insufficient inductive bias would result in misclassification of examples from an in-distribution test set as OOD. Labeled data provide this additional information, as relevant factors can be defined as those that help the classification task, with the limitation that there might be more factors involved in characterising the in-distribution than those needed to predict the labels.

In this work, we introduce a general framework for OOD detection problems that does not require label information. Our framework can be widely applied to OOD detection tasks, including visual, audio, and textual data with the only limitation that transformations must be *a priori* known that conserve the semantics of training examples, such as geometric transformations for images, proximity of time intervals for audio recordings (van den Oord et al. (2018)), or randomly masking a small fraction of words in a sentence or paragraph (Devlin et al. (2019)). For visual data we show new state-of-the-art OOD classification accuracies for standard benchmark data sets, surpassing even the accuracies that include labels as additional information. The key contributions of this work are

- We propose a new OOD detection framework that is applicable in absence of labeled in-distribution data or OOD examples that are labeled as such.
- We show that our approach strongly improves OOD detection for challenging tasks in the visual domain
- We find that identifying semantically close examples in the training set is central for reliable OOD detection

## 2 RELATED WORK

**Unsupervised Methods using in-distribution labels**. Many OOD detection methods make use of labels to generate scores that are either based on class prediction probabilities or on intermediate representations for an in-distribution classification task. For example, Hendrycks & Gimpel (Hendrycks & Gimpel (2017)) used the maximum of the softmax probabilities (MSP) to discriminate between OOD and in-distribution. More recent approaches Lee et al. (2018a); Winkens et al. (2020); Zhang et al. (2020) use labels to learn an intermediate representation on which a density distribution (e.g. multivariate normal distribution or deep generative network) can be fitted, which then can be used to compute the likelihood of OOD examples. As labels implicitly provide information about the semantic relation of examples in the training set, approaches using label information typically show higher accuracy than unsupervised methods. These approaches can be improved by introducing additional parameters or training strategies. For example, MSP was improved by introducing a temperature parameter (Liang et al. (2018)), alternative losses (Lee et al. (2018a); Vyas et al. (2018)), auxiliary objectives (Devries & Taylor (2018); Hendrycks et al. (2019b); Mohseni et al. (2020)), or outlier exposure (Hendrycks et al. (2019a)). Intermediate representations were improved using a multi-head network architecture (Shalev et al. (2018), contrastive learning Winkens et al. (2020), metric learning Masana et al. (2018)).

**General Unsupervised Methods**. If label information is absent, other means must be found to impose an inductive bias on the OOD detection model to generalise over the training set. Existing approaches can be separated in methods that learn generalisable features based on (i) self-supervised learning tasks Golan & El-Yaniv (2018), transformations that destroy semantics Choi & Chung (2020), match of encoder-decoder architectures Xiao et al. (2020), or make use of a semantically related auxiliary outlier distribution Schirrmeister et al. (2020). The work that is most related to ours is Geometric-Transformation Classification (GEOM), proposed by Golan & El-Yaniv (2018)

and improved by Bergman & Hoshen (2020), which belongs to the class of self-supervised learning approaches (Hendrycks et al. (2019b)). The central idea of GEOM is to construct an auxiliary in-distribution classification task by transforming each image of the training set by one of 72 different combinations of geometric transformations with fixed strength, such as rotation, reflection, and translation. The task is to predict which of the 72 transformations has been applied, given a transformed image. GEOM gives examples that show high prediction uncertainty a high OOD score. The relevant features learned by this task are salient geometrical features, such as the typical orientation of an object. Our approach differs from GEOM by the fact that we define the relevant features as those that are invariant under geometric and other transformations, such as cropping and color jitter, which are chosen of moderate strength to not change the semantics of the images in the training set.

## 3 METHOD

An intuitive approach for OOD detection is to learn a representation that densely maps the in-distribution to a small region within a lower dimensional space (latent space), with the consequence that OOD examples will be found outside this region with high probability. The representation should include the salient semantic information of the training set, to ensure that test examples from the in-distribution are not misclassified as OOD, but disregard irrelevant factors of variation that would prevent dense mapping. As learning this mapping by Autoencoders is difficult, we split the OOD detection task into finding a semantically dense mapping of in-distribution onto a $d$-dimensional unit-hypersphere by contrastive learning, followed by classifying neighbouring examples on the unit-hypersphere as semantically close or distant.

### 3.1 LEARNING SEMANTIC SIMILARITY

A contrastive objective can be used to align feature vectors $h(x) \in \mathbb{R}^d$ that are semantically similar and at the same time distributes examples of the training set almost uniformly over the unit-hypersphere (Wang & Isola (2020); Chen et al. (2020)). This representation allows to identify for any test example the semantically close example from the training set. The mapping $h(x) = f(x)/\|f(x)\|$ can be learned from training a deep neural network $f(x)$ to minimise the contrastive loss

$$\mathcal{L}[h] = -\mathbb{E}_{(x,x')\sim\mathcal{T}_h(x,x')}\left[\log \frac{e^{h(x)^T h(x')/\tau}}{\mathbb{E}_{x_{neg}\sim\mathcal{T}_h(x)}\left[e^{h(x)^T h(x_{neg})/\tau}\right]}\right] \qquad , \qquad (1)$$

where $\tau$ denotes a temperature parameter. Here, each positive pair $(x, x')$ is the result of sampling from a distribution of transformations $\mathcal{T}_h(x, x')$ that conserve semantics between $x$ and $x'$, with $\mathcal{T}_h(x')$ the marginal of $\mathcal{T}_h(x, x')$. For datasets used to benchmark object recognition tasks, samples $(x, x') \sim \mathcal{T}_h(x, x')$ can be generated by picking a single example from the training set and independently apply random transformations, such as geometric transformations, colour distortions, or cropping (Appendix D). The negative pairs can be generated by applying random transformations to different training examples. We emphasise that the types of transformations and their strengths essentially define the semantics we want to encode and thus determine if, for example, the image of a black swan is classified as OOD for an in-distribution that contains only white swans. The design of transformations that capture the underlying semantics of the training dataset requires either higher level understanding of the data or extensive sampling of different combinations of transformations with evaluation on an in-distribution validation set.

### 3.2 LEARNING SEMANTIC DIFFERENCES

As the encoder $h(x)$ maps any example $x$ on the unit-hypersphere, including OOD examples, we have the situation that OOD examples can be close to training examples without sharing semantic information. The obvious reason is that the contrastive objective homogeneously distributes training examples on unit-hypersphere, giving no preferred direction for OOD examples to cluster. As a consequence, the representation $h(x)$ cannot be directly used for OOD detection. The idea is now to make use of the fact that nearby examples on unit-hypersphere share semantic information if both come from the in-distribution but don't share semantic information if one of the two examples is

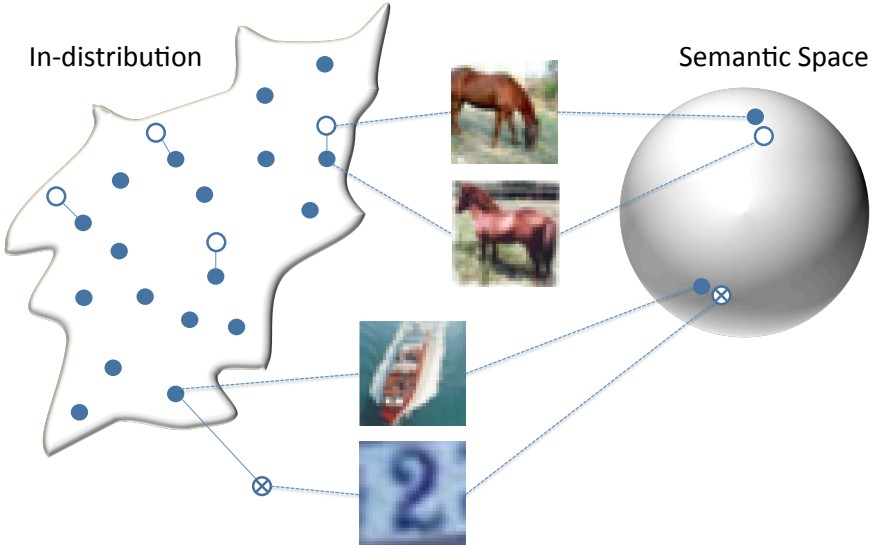

Samples : ● Training ○ Test(in-distribution) ⊗ Test (OOD)

Figure 1: Illustration of mapping examples from the in-distribution onto a unit-hypersphere. In this representation, feature vectors from the in-distribution are semantically similar if they approximately align and semantically diverse if they are separated by a large angle. If OOD examples are mapped onto the unit-hypersphere, they can align with training examples without being semantically similar. A discriminator trained to classify pairs of feature vectors from the training set as semantically close or distant, with sharing or not sharing the same semantic neighbourhood as target, can then be used for detecting OOD examples.

OOD. We therefore train a score function $s(x, x')$ to detect semantic differences between nearby examples on the unit-hypersphere, which are determined by the readily trained encoder $h(x)$. A statistically meaningful score can be given by the likelihood ratio between the probability $\mathcal{P}_{pos}$ of a test example $x_{test}$ and its next nearest neighbour in the training set $x_{next} = \arg\max_x h^T(x)h(x_{test})$ to be semantically close in relation to the likelihood $\mathcal{P}_{neg}$ that the same two examples are semantically distant

$$s^*(x_{next}, x_{test}) = \log \frac{\mathcal{P}_{pos}(x_{next}, x_{test})}{\mathcal{P}_{neg}(x_{next}, x_{test})} \qquad (2)$$

For the distribution of positive examples we use $\mathcal{P}_{pos}(x, x') = (1 - z)\mathcal{T}_p(x, x') + z\mathbf{1}_{x' \in \mathbb{S}(x)}/|\mathbb{S}(x)|$, with $z$ a Bernoulli distributed random variable of mean $\mu$. We introduced with $\mathbb{S}(x)$ the semantic neighbourhood of $x$, which is defined by the $k$-nearest neighbours of $x$, using cosine similarity $h^T(x)h(x')$ as semantic similarity measure (Fig. 2). The type of transformations are similar to $\mathcal{T}_h$ but with reduced strength to ensure that the relevant factors of variation are conserved (Fig. 3 and Appendix D). For negative examples we take $\mathcal{P}_{neg}(x, x') = \mathcal{T}_n(x')\mathcal{T}_n(x)$, with $\mathcal{T}_n(x)$ the marginals, which implies that $x$ and $x'$ are almost always derived from two different images of the training set. The negative transformations, $\mathcal{T}_n$, are allowed to include stronger and more diverse transformations than $\mathcal{T}_p$ (Fig. 3 and Appendix D). In principle negative examples can be constructed that are harder to classify, such as augmenting $\mathcal{P}_{neg}$ by pairs that are independent transformations of the same example (Choi & Chung (2020)). However, we found that this reduces the performance (Table 2). As shown in Appendix A, the score $s^*(x, x')$ maximises the training objective $J[s; \gamma]$, given by

$$J[s; \gamma] \quad = \quad \mathbb{E}_{(x,x')\sim\mathcal{P}_{pos}}\Big[\log \sigma(a)\Big] + \gamma\mathbb{E}_{(x,x')\sim\mathcal{P}_{neg}}\Big[\log\big(1 - \sigma(a)\big)\Big] \qquad (3)$$

Here, we defined $\sigma(a) = 1/(1 + e^{-a})$ and $a = s(x, x') - \log(\gamma)$ and realise the score function $s(x, x')$ by a deep neural network. It can be shown that the optimal solution $s^*(x, x')$ is invariant

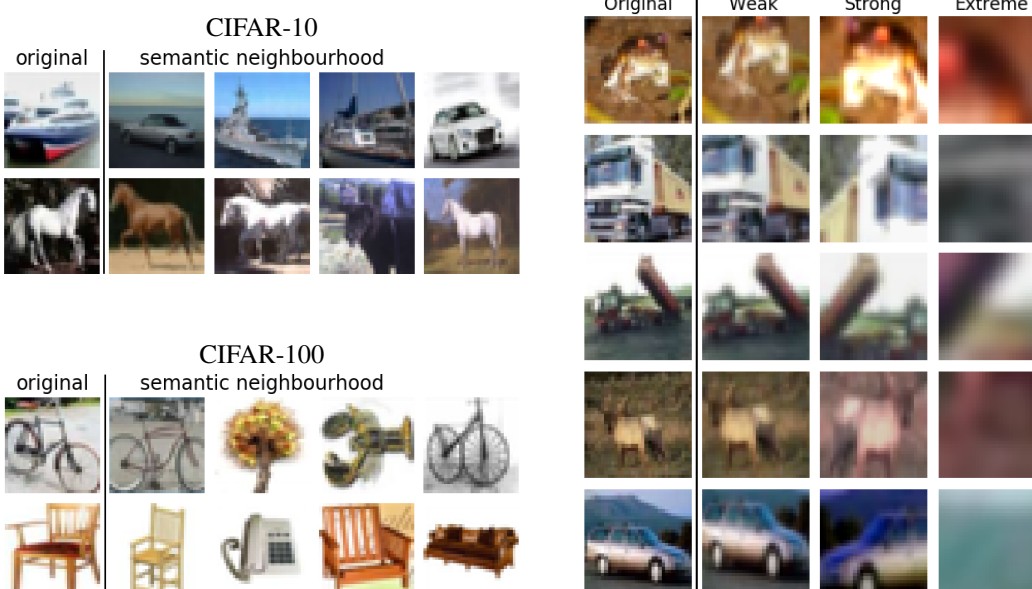

Figure 2: Semantic Neighbourhoods for examples from CIFAR-10/100.

Figure 3: Transformation strengths used in training $P_{pos}$ (weak to strong) and $P_{neg}$ (weak to extreme)

to any variation in $\gamma > 0$ (Appendix A). We introduced $\gamma$ as it is notoriously hard to learn ratios of probability densities in high dimensional spaces, which is a central problem of generative adversarial networks (Azadi et al. (2019)). In general, $s(x, x')$ learned by the objective Eq. 3 can deviate significantly from the optimal generalising likelihood ratio $s^*(x, x')$. This deviation is most apparent if $P_{pos}(x, x')$ is close to zero where $P_{neg}(x, x')$ is non-zero and vice versa, as shown in Fig. 4. In this case the objective can be maximised by any decision boundary that lies in the region between the distributions $P_{pos}(x, x')$ and $P_{neg}(x, x')$. To smoothen the score function $s(x, x')$ we sample $\gamma$ at each iteration of the learning process and thereby effectively sample over an ensemble of gradients (Appendix B). Inspired by the lottery ticket hypothesis that training a deep neural network under constant objective mainly affects the weights of a small subnetwork (Frankle & Carbin (2019)), we can reason that sampling over $\gamma$ affects the weights for an ensemble of overlapping subnetworks. As a consequence, $s(x, x')$ is the prediction from an ensemble of models, which typically results in higher prediction accuracies, less variance with respect to weight initialisation, and higher robustness to overfitting. The effect of uniform sampling of $\gamma$ on stabilising the decision boundary and thus observing the train/test sets from the in-distribution within the positive score range is shown in Appendix C. Although only the difference in score values between a test example and the in-distribution test set is relevant for OOD detection, examples from the in-distribution should be sufficiently distant from $P_{neg}(x, x')$ for optimal performance. It can be further shown that for the extreme case $\gamma \to \infty$ the score function learns the optimal weights to realise importance sampling for $P_{pos}$ by sampling from $P_{neg}$ (Appendix B).

## 4 TRAINING AND EVALUATION PROTOCOLS

### 4.1 TRAINING

Experiments were carried out using either ResNet18 or ResNet34 neural network architectures, for both the encoder $f(x)$ and the discriminator $s(x, x')$. We substituted ReLU activations for the discriminator by ELU units to reduce the overconfidence induced by ReLU (Meinke & Hein (2020)), which resulted in a strong reduction of unusually large spikes in the training loss curve for the discriminator. For contrastive learning a MLP head was attached to the last hidden layer that projects to a $d = 128$ dimensional feature vector, $h(x)$, whereas for the discriminator the MLP head projects

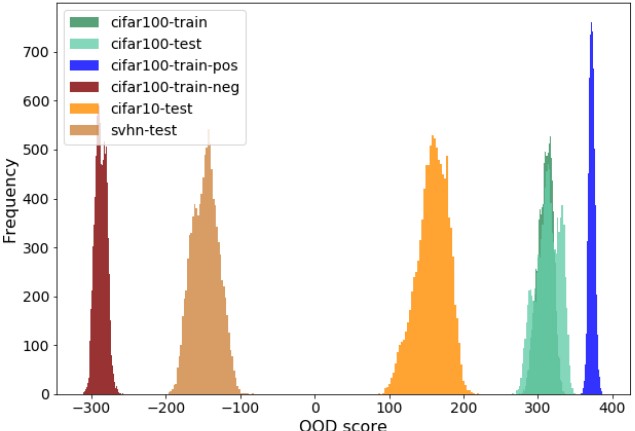

Figure 4: Distributions over the OOD detection score, $s(x, x')$, trained on CIFAR-100 pos/neg pairs ($P_{pos}$ in blue; $P_{neg}$ in red) as described in Section 4.1 and applied to semantic nearest-neighbour pairs from the test sets of SVHN and CIFAR-10 (out-distributions) in comparison to semantic nearest-neighbour pairs of the CIFAR-100 test/train sets (in-distributions).

to a scalar output, $s(x, x')$. Note that the ResNets for encoder and discriminator don't share parameters. We train the contrastive loss at batch size $2048$ and the discriminator at batch size $128$, using ADAM(AMSGrad) optimiser. We applied random transformations to each example in the training set before presenting them to encoder and discriminator. The transformations consist of combinations of *random cropping* followed by resizing to the original size, *random horizontal flipping*, *random color jitter*, *grey scaling*, and *gaussian blurring* (Appendix D). For training the encoder, $h(x)$, we used the same transformations with the same strength as reported in Chen et al. (2020)). We set the temperature parameter to $\tau = 1$, which is the value reported in Winkens et al. (2020)) for the same datasets used in this work. Unless otherwise specified, positive pairs for training the discriminator are two independent transformations of a single image from the training set, where the transformation strength is bounded by *strong* transformations (Fig. 3 and Appendix D), to make sure that we don't transform out of the in-distribution. As pairs generated from independent transformations of the same image are typically semantically closer than any two semantically close images of the in-distribution test set (Fig. 2), the latter would be erroneously classified as OOD. To avoid mis-classification we augment the transformed positive pairs with a fraction of semantic similar pairs, with pairing partners randomly selected from the semantic neighbourhood. The strength of augmentation is determined such that train/test sets from the in-distribution reside on the positive OOD-score side yet remain inside the sensitive range of the logistic sigmoid function (Fig. 4). Unless otherwise specified, we take a semantic neighbourhood size of 4 and substitute a fraction $\mu = 1/32$ of transformed pairs in a minibatch with semantically similar pairs from the training set. For regularisation, we use weight decay of $10^{-6}$ and uniformly sample $\gamma \sim \mathcal{U}(1, 10)$ at each iteration of the learning process. Negative pairs are constructed by transforming two different examples from the training set, including also 'extreme' transformations and gaussian blur.

## 4.2 EVALUATION

We evaluate the results using Area Under the Receiver Operating Characteristic curve (AUROC), which has the advantage to be scale-invariant – measures how well predictions are ranked, rather than their absolute values – and classification-threshold-invariant – it measures how well OOD samples are separated from the training set. However, for any practical setting of OOD detection a classification threshold is needed and can be chosen such that the false positive rate of an in-distribution test set is close to some threshold, e.g. $\alpha = 0.05$. We do not report values for Area Under the Precision-Recall curve (AUPR) as in this work we have no class imbalance between the OOD test set and the in-distribution test-set. As we observed significant shifts of OOD-scores for in-distribution

Table 1: Out-of-distribution detection performance (% AUROC). Reported values for SemSAD are lower bounds.

| Method | $\mathcal{D}_{in}$ : CIFAR10
$\mathcal{D}_{out}$ : SVHN | CIFAR10
CIFAR100 | CIFAR100
CIFAR10 |
|---|---|---|---|
| *Unsupervised methods using in-distribution labels* | | | |
| Softmax probs. (Hendrycks & Gimpel, 2017) | 89.9 | 86.4 | 77.1 |
| ODIN(Liang et al., 2018) | 96.7 | 85.8 | 77.2 |
| Mahalanobis (Lee et al., 2018b) | 99.1 | 88.2 | 77.5 |
| Residual flows (Zisselman & Tamar, 2020) | 99.1 | 89.4 | 77.1 |
| Outlier exposure (Hendrycks et al., 2019a) | 98.4 | 93.3 | 75.7 |
| Rotation pred. (Hendrycks et al., 2019b) | 98.9 | 90.9 | - |
| Gram matrix (Sastry & Oore, 2020) | 99.5 | 79.0 | 67.9 |
| Contrastive Aug. (Winkens et al., 2020) | 99.5 | 92.9 | 78.3 |
| OpenHybrid (Zhang et al., 2020) | 99.8 | 95.1 | 85.6 |
| *General unsupervised methods* | | | |
| Likelihood Regret (Xiao et al., 2020) | 86.6 | - | - |
| SVD-RND (Choi & Chung, 2020) | 96.9 | - | - |
| Hierachical-AD (Schirrmeister et al., 2020) | 99.0 | 86.8 | 62.5 |
| **SemSAD (Ours)** | **100** | **99.9** | **99.9** |

train/test sets between training runs (Appendix C), we suggest for any practical applications to carry out a majority vote over 5 independent training runs, where after each run an example is classified as OOD if the OOD-score is significantly lower than the OOD-scores of the in-distribution test set.

## 5 EXPERIMENTAL RESULTS

In our experiments, we focus on difficult OOD detection problems in the visual domain (Nalisnick et al. (2019)), in particular CIFAR-10/SVHN, CIFAR-10/CIFAR-100, and CIFAR-100/CIFAR-10. The main results are summarised in Table 1, where we used an identical setup (e.g. same hyperparameters, transformation strengths, and network size) for all datasets and averaged AUROC values over 5 subsequent runs using ResNet18 and 5 subsequent runs using ResNet34. We compare our unsupervised method (SemSAD) with supervised methods that use label information and/or OOD examples that are labeled as such. Surprisingly, we find that SemSAD outperforms not only all unsupervised methods on these problems but also all supervised methods we are aware of. This result is especially striking as supervised methods typically outperform unsupervised methods, as semantic representations learned on label information are typically of advantage. Our interpretation is that learning a representation for the semantic information shared between pairs of examples allows to identify a larger set of relevant features that characterise the in-distribution than from the large number of examples that share the same label. The more of the relevant features can be identified the tighter the in-distribution can be characterised, which helps OOD detection. The performance gain of our method is strong for all OOD detection problems considered in this work but most apparent for CIFAR-100 as in-distribution and OOD examples from CIFAR-10, with increase in state-of-the-art AUROC for unsupervised methods not using label information from 0.625 to 0.999. Note that the classes of CIFAR-10 and CIFAR-100 are mutually exclusive, and thus CIFAR-10 can be used as OOD test set.

We carried out further experiments to see the effects of hyperparameter values and the transformations used for training the discriminator (Table 2). As expected, if we destroy semantic information by using extreme transformations for generating positive (in-distribution) pairs the performance is significantly reduced, whereas Gaussian blur on negative examples has a positive effect. The experiments further show that the semantic neighbourhood size should be taken small enough to make sure that the pairs generated from semantic neighbourhoods and used for training $s(x, x')$ are semantically close. In general, we observed better performance if we broaden the distribution of negative pairs, e.g. by augmenting examples with gaussian blur and use extreme transformations. The per-

Table 2: Average over AUROC values from 5 independent training runs for CIFAR-100/CIFAR10 (in/out distribution) for different setups. The lowest AUROC values among the 5 runs are shown in brackets. Reported AUROC values are lower bounds. We applied gaussian blurring on negative samples (blur), extreme transformations on positive samples (extreme transf.), and using correlated negative pairs $P_{neg}(x, x')$ derived from extreme transformations of the same image (correlated neg), and changed the fraction of semantically similar pairs ($\mu$) per minibatch, the sampling range for $\gamma$, and the semantic neighbourhood size (N). AUROC is computed for CIFAR-100/10 test sets with 10k examples.

| AUROC | blur | extreme transf. | $\mu$ | $\gamma$ | N | correlated neg |
|---|---|---|---|---|---|---|
| 0.980 (0.915) | - | ✓ | 1/32 | $\sim \mathcal{U}[1, 10]$ | 4 | - |
| 0.999 (0.999) | ✓ | - | 1/32 | $\sim \mathcal{U}[1, 10]$ | 4 | - |
| 0.999 (0.999) | ✓ | - | 1/32 | 1 | 4 | - |
| 0.991 (0.963) | ✓ | - | 1/32 | $\sim \mathcal{U}[1, 100]$ | 4 | - |
| 0.998 (0.996) | ✓ | - | 1/32 | $\sim \mathcal{U}[1, 10]$ | 32 | - |
| 0.729 (0.564) | ✓ | - | 1/32 | $\sim \mathcal{U}[1, 10]$ | 4 | ✓ |
| 0.999 (0.999) | ✓ | - | 0 | $\sim \mathcal{U}[1, 10]$ | 4 | - |

formance of our method is tightly connected to the ability of the encoder $h(x)$ and discriminator $s(x)$ to extract or be sensitive to features that allow to generalise over the training-set and are thus specific to the in-distribution. These generalisable features are orthogonal to the features that change under the transformations we use in training. The type of transformations we use in this work, e.g. cropping, horizontal flip, and colour jitter, are generic in the sense that they are designed to conserve the semantics for images that are related to physical objects. In general, the type of transformations used must match the generalisable features of the in-distribution. For example, if all examples of in-distribution have a specific horizontal orientation, then horizontal flip must be excluded from the transformations.

## 6 CONCLUSION

In this work we proposed SemSAD – a new OOD detection approach that is based on scoring semantically similar pairs of examples from the in-distribution. We showed that our method outperforms supervised and unsupervised methods on challenging OOD detection tasks in the visual domain. The definition of semantic similarity within our approach requires to identify transformations that are applicable to individual examples and are orthogonal to the salient semantic factors of the in-distribution. Although semantic similarity can be broadly defined as "everything that is not noise", high predictive power can be expected if the semantic similarity score catches the higher order features that are specific to the in-distribution. In practice, there are problems where the definition of semantic similarity is challenging. For example, if genome sequence data is the input and the effect on the phenotype of the organism is the true underlying semantics. In this case, it is unclear how transformations of the genome sequence should look like that lead to the same phenotype. In contrast, for the important problem of protein folding, such transformations can be inferred from multiple sequence alignments of sequences that likely conserved the function of a protein.

AUTHOR CONTRIBUTIONS

ACKNOWLEDGMENTS

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

## A   APPENDIX

The objective

$$J[s; \gamma] \quad = \quad \gamma_{pos} \mathbb{E}_{(x,x') \sim \mathcal{P}_{pos}} \Big[ \log \sigma(a) \Big] + \gamma_{neg} \mathbb{E}_{(x,x') \sim \mathcal{P}_{neg}} \Big[ \log \big( 1 - \sigma(a) \big) \Big] \tag{4}$$

with $\sigma(a) = 1/(1 + e^{-a})$, $a = s(x, x') + \log(\gamma_{pos}/\gamma_{neg}))$, and $\gamma = (\gamma_{pos}, \gamma_{neg})$, has the upper bound $J[s^*; \gamma] \geq J[s; \gamma]$ for all $\gamma_{pos}, \gamma_{neg} > 0$, where $s^*$ is given by

$$s^*(x, x') = \log \frac{\mathcal{P}_{pos}(x, x')}{\mathcal{P}_{neg}(x, x')} \tag{5}$$

under the condition that $P_{pos}$ and $P_{neg}$ have the same support – that is where $P_{pos}$ is non-zero also $P_{neg}$ is non-zero and vice versa.

To prove that assertion we make use of variational calculus (see e.g. C. Bishop, Patter Recognition and ML, Appendix D) to compute the functional derivative $\delta J/\delta s$, which is defined by the integral over an arbitrary test function, $\eta(x, x')$,

$$\frac{J[s + \epsilon \eta; \gamma]}{d\epsilon} \bigg|_{\epsilon=0} \quad = \quad \int \frac{\delta J[s; \gamma]}{\delta s(y, y')} \eta(y, y') dy dy' \tag{6}$$

$$= \quad \int \Big[ \gamma_{pos} \mathcal{P}_{pos}(x, x') \big( 1 - \sigma(a) \big) - \gamma_{neg} \mathcal{P}_{neg}(x, x') \sigma(a) \Big] \eta(x, x') dx dx' \tag{7}$$

where we have used that $d\sigma(a)/ds = \sigma(a)\big(1 - \sigma(a)\big)$. The optimum can be computed from $\delta J/\delta s|_{s=s^*} = 0$, which results in

$$\gamma_{pos} \mathcal{P}_{pos}(x, x') \big( 1 - \sigma(a) \big) - \gamma_{neg} \mathcal{P}_{neg}(x, x') \sigma(a) = 0 \tag{8}$$

$$\Rightarrow \quad \frac{\gamma_{pos} \mathcal{P}_{pos}(x, x')}{\gamma_{pos} \mathcal{P}_{pos}(x, x') + \gamma_{neg} \mathcal{P}_{neg}(x, x')} = \frac{1}{1 + e^{-s^*(x,x') - \log(\gamma_{pos}/\gamma_{neg})}} \tag{9}$$

$$\Rightarrow \quad s^*(x, x') = \log \frac{\mathcal{P}_{pos}(x, x')}{\mathcal{P}_{neg}(x, x')} \qquad \forall \gamma_{pos}, \gamma_{neg} > 0 \tag{10}$$

Note that $J[s; \gamma]$ is not bounded from below, so the optimum is a maximum.

# B    Appendix

We show that optimising the objective $J[s; \gamma_{pos}, \gamma_{neg}]$ by gradient ascent, with $\gamma_{pos}, \gamma_{neg} > 0$ randomly sampled in each optimisation step, leads to averaging over an ensemble of gradients. A gradient based optimisation method in its simplest form updates the parameters $\theta$, which determine the function $s(x, x')$, by the rule

$$\theta \leftarrow \theta + \alpha \nabla_\theta J[s; \gamma] \tag{11}$$

with $\alpha$ the learning rate and

$$
\begin{aligned}
\nabla_\theta J[s; \gamma] &= \gamma_{pos} \mathbb{E}_{(x,x') \sim \mathcal{P}_{pos}} \left[ \big( 1 - \sigma(a) \big) \nabla_\theta s(x, x') \right] - \gamma_{neg} \mathbb{E}_{(x,x') \sim \mathcal{P}_{neg}} \left[ \sigma(a) \nabla_\theta s(x, x') \right] \\
&= \mathbb{E}_{(x,x') \sim \mathcal{P}_{pos}} \left[ \frac{\gamma_{pos}}{1 + \frac{\gamma_{pos}}{\gamma_{neg}} e^{s(x,x')}} \nabla_\theta s(x, x') \right] \\
&\quad - \mathbb{E}_{(x,x') \sim \mathcal{P}_{neg}} \left[ \frac{\gamma_{neg}}{1 + \frac{\gamma_{neg}}{\gamma_{pos}} e^{-s(x,x')}} \nabla_\theta s(x, x') \right]
\end{aligned}
\tag{12}
$$

This result shows that for given $s(x, x')$, random values of $\gamma_{pos}, \gamma_{neg} > 0$ weight the expected gradients for the positive examples and for the negative examples differently. As a consequence, $\nabla_\theta J[s; \gamma]$ takes different directions for each parameter update, given fixed (mini-)batch and fixed initial conditions.

If we consider the following limiting cases for $s \approx s^*$

$$
\begin{aligned}
\lim_{\substack{\gamma_{pos} \to 1 \\ \gamma_{neg} \to \infty}} \nabla_\theta J[s; \gamma] &= \mathbb{E}_{(x,x') \sim \mathcal{P}_{pos}} \left[ \nabla_\theta s(x, x') \right] - \mathbb{E}_{(x,x') \sim \mathcal{P}_{neg}} \left[ e^{s(x,x')} \nabla_\theta s(x, x') \right] \tag{13} \\
&\approx \mathbb{E}_{(x,x') \sim \mathcal{P}_{pos}} \left[ \nabla_\theta s(x, x') \right] - \mathbb{E}_{(x,x') \sim \mathcal{P}_{neg}} \left[ \frac{\mathcal{P}_{pos}(x, x')}{\mathcal{P}_{neg}(x, x')} \nabla_\theta s(x, x') \right] \tag{14}
\end{aligned}
$$

and

$$
\begin{aligned}
\lim_{\substack{\gamma_{pos} \to \infty \\ \gamma_{neg} \to 1}} \nabla_\theta J[s; \gamma] &= \mathbb{E}_{(x,x') \sim \mathcal{P}_{pos}} \left[ e^{-s(x,x')} \nabla_\theta s(x, x') \right] - \mathbb{E}_{(x,x') \sim \mathcal{P}_{neg}} \left[ \nabla_\theta s(x, x') \right] \tag{15} \\
&\approx \mathbb{E}_{(x,x') \sim \mathcal{P}_{pos}} \left[ \frac{\mathcal{P}_{neg}(x, x')}{\mathcal{P}_{pos}(x, x')} \nabla_\theta s(x, x') \right] - \mathbb{E}_{(x,x') \sim \mathcal{P}_{neg}} \left[ \nabla_\theta s(x, x') \right] \tag{16}
\end{aligned}
$$

we see that the objective learns the optimal importance weights of importance sampling. As in this work we have the situation that $\mathcal{P}_{pos}(x, x') = 0$ for some negative examples, the case $\gamma_{pos} \to \infty, \gamma_{neg} \to 1$ should not be applied, which is why we set $\gamma_{pos} = 1$ and sample uniformly from $\gamma_{neg} \in [1, N]$, with $N > 1$.

## C APPENDIX

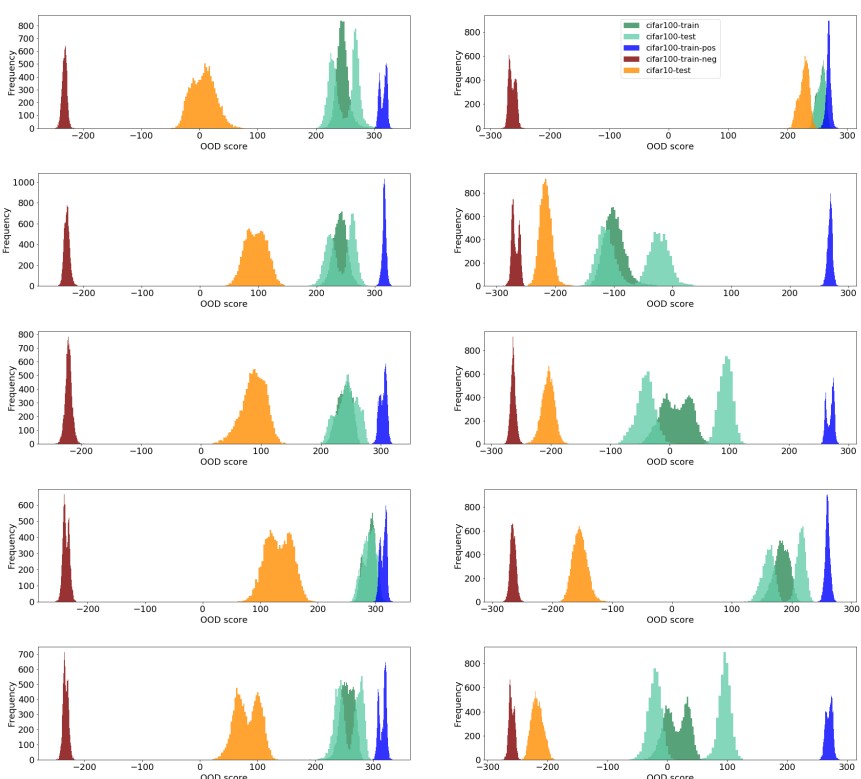

Figure 5: Random shifts of in-distribution test/train sets (green) for different training runs as indicator for instability of the decision boundary. Shown are results for 5 independent training runs for $\gamma \sim \mathcal{U}(1, 10)$ (left column) and for $\gamma = 1$ (right column), using the same setup as used to compute Table 1 but with ResNet18.

# D  APPENDIX

## D.1  TRAINING CONTRASTIVE ENCODER

In order to train the contrastive encoder, we use a modified version of resnet18 with 128-dimensional projection head to make it suitable for cifar10 and cifar100 datasets with relatively small image size. In particular we remove the Maxpooling layer and subsitute the first $7 \times 7$ convolutional layer of stride 2 with a convolutional layer of kernel size $3 \times 3$ with padding and stride of 1. For the optimisation task, we use the Adam optimiser with learning rate of $3 \cdot 10^{-4}$ and weight decay of $10^{-6}$. The network is trained for 1500 epochs at batch size 2048.

## D.2  TRAINING DISCRIMINATOR

To train the discriminator, we use ResNet18/34 and apply the same modifications as for the contrastive encoder. In addition, all the ReLU activation functions are replaced with ELU and the projecting head maps to a scalar value. Note that encoder and discriminator don't share parameters. The discriminator is trained with initial learning rate of $5 \cdot 10^{-5}$ using AMSGrad optimiser with weight decay of $10^{-6}$ on batch size of 128 samples in each iteration. The learning rate is multiplied by 0.2 and 0.1 after 200 and 500 epochs, respectively. To generate the positive pairs for training, we first find the 4 examples with the highest cosine similarity score among 10k random examples for each example in training set, from which one is randomly selected with equal chance. During the training procedure $\mu = 1/32 = 3.125\%$ of each batch includes semantically similar pairs. For the $\gamma$, in each iteration a value is uniformly chosen from the range $\mathcal{U}(1, 10)$. The hyperparameters and their default values are shown in Table 3

Table 3: Hyperparameters for discriminator

| Hyperparameter | Default Value |
|---|---|
| Batch size | 128 |
| Initial learning rate | $5 \cdot 10^{-5}$ |
| Portion of semantic samples | 3.125% |
| Size of neighbourhood | 4 |
| Number of samples for best neighbour selection | $10k$ |
| $\gamma$ | $\sim \mathcal{U}(1, 10)$ |
| Weight decay | $10^{-6}$ |

## D.3  GEOMETRIC TRANSFORMATIONS

As data augmentation to train the contrastive encoder, we use the same transformations as in Chen et al. (2020). including, randomly chosen geometric transformation from the set {Cropping, Horizontal Flip, Color Jitter, GrayScale, Gaussian Blurring}. Pytorch snippets for encoder transformations can be found in Table 4. To train the discriminator we make two sets of transformations, one for positive samples and one for negative ones. The main intuition to shape a set of transformation for positive samples is to keep them in-distribution according to the original training samples. For the special case of cropping we make three different categories as weak, strong and extreme cases. Table 5 shows their cropping scale according to pytorch standard. To make positive pairs we randomly apply both weak and strong ranges for cropping, random horizontal flipping and color jittering on the same image from the training set and for negative pairs we apply all ranges of weak, strong and extreme cropping, horizontal flipping, color jittering, and Gaussian blurring on two randomly selected images. The details and pytorch snippet for positive and negative transformations can be found in Table 7 and 6 respectively. Note that for augmentation with semantically similar pairs from the training set there is no transformation applied on positive pairs.

Table 4: Transformations for contrastive encoder

| Transformation | Pytorch snippet |
| --- | --- |
| Cropping | transforms.RandomResizedCrop(32 , scale=$(0.08, 1)$) |
| Horizontal Flip | transforms.RandomHorizontalFlip($p = 0.5$) |
| Color Jitter | transforms.RandomApply([ transforms.ColorJitter($0.4, 0.4, 0.4, 0.1$ )], p=0.8) |
| GrayScale | transforms.RandomGrayscale($p = 0.2$) |

Table 5: Weak, strong and extreme cropping

| Strenght | Pytorch snippet |
| --- | --- |
| Weak | transforms.RandomResizedCrop(32 , scale=$(0.7, 1)$) |
| Strong | transforms.RandomResizedCrop(32 , scale=$(0.3, 0.7)$) |
| Extreme | transforms.RandomResizedCrop(32 , scale=$(0.08, 0.3)$) |

Table 6: Transformations for negative samples

| Transformation | Pytorch snippet |
| --- | --- |
| Cropping | transforms.RandomResizedCrop(32 , scale=$(0.08, 1)$) |
| Horizontal Flip | transforms.RandomHorizontalFlip($p = 0.5$) |
| Color Jitter | transforms.RandomApply([ transforms.ColorJitter($0.4, 0.4, 0.4, 0.1$ )], p=0.8) |
| Gaussian Blurring | GaussianBlur($\sigma \sim \mathcal{U}(0.1, 2.0)$, $p = 0.2$) link to github |

Table 7: Transformations for positive samples

| Transformation | Pytorch snippet |
| --- | --- |
| Cropping | transforms.RandomResizedCrop(32 , scale=$(0.4, 1)$) |
| Horizontal Flip | transforms.RandomHorizontalFlip($p = 0.5$) |
| Color Jitter | transforms.RandomApply([ transforms.ColorJitter($0.4, 0.4, 0.4, 0.1$ )], p=0.8) |

