# OpenReview forum: "UNSUPERVISED ANOMALY DETECTION FROM SEMANTIC SIMILARITY SCORES"
_ICLR.cc/2021/Conference — Reject_

### Official Review · AnonReviewer1 · 2020-10-28
**A dimensionality reduction based approach to anomaly detection for images that tries to overcome certain disadvantages of autoencoders**

**Rating:** 5
**Confidence:** 3

**Review:**

The paper proposes an OOD detector algorithm that first learns a function to reduce the data dimensionality followed by learning a classifier discrimination model to separate in-distribution data from OOD.

Pro:
1. The paper compares many baseline algorithms
2. The paper tries to address an important problem (OOD detector)

Con:
1. The paper title is 'A General Framework...', however, the few datasets selected for experiments represent a very narrow domain. The paper title should be narrowed down or more domains should be included in experiments.
2. There are gaps in the intuitions such as why would two instances in the same neighborhood in the reduced dimension not be expected to have similar labels.

Main Comments:

1. The overall approach is that of reducing the dimensionality of the data by projecting it onto a lower dimensional manifold (surface of hyper sphere) and then using a discriminator. This approach is not novel in general.

2. While the paper claims that this is a general technique, it depends on the concept of 'semantic neighborhood' for which it only provides CIFAR variants as evidence. We do not know (contrary to claims) whether it might work on other types of data (audio, text, etc.)

3. Section 4: "Our interpretation ... includes all semantic information ... helps OOD detection. In contrast, learning from label information ... mainly the semantics that help predicting labels." -- The paper does admit that the 'semantic neighborhood' is ill defined (Section 6, Conclusion). Yet the paper assumes, in Section 4, that the proposed technique (using pairwise distance metric) learns it well for the image data it was tested on. It is hard to see how this interpretation is justified. My assumption is that the algorithm has only learned what is necessary for the task of OOD just as a classification algorithm will learn what is necessary for labeling. There are many critical decisions that have gone in to design the proposed OOD detector (such as the distance metric to use, which features to use for the discriminator, etc.). It is more conceivable that in the end the algorithm has learned just enough representation that makes the combined design choices work well on the specific dataset. It is hard to generalize given that the experiments cover so few datasets. I suggest the paper remove 'semantic neighborhood' terminology.

4. Section 2.2: "...belief in the lottery hypothesis..." -- Many of the subnetworks might be sharing weights and are therefore not independent. This point becomes more important because as discussed in Section 3.1, a small network was used which increases the likelihood of weight-sharing. So, the true ensemble effect might be absent in reality.

5. Section 2.2: "The idea is now to make use of the fact that nearby examples on unit-hypersphere share semantic information if both come from the in-distribution but don’t share semantic information if one of the two examples is OOD." -- It is not clear to me why any two close examples would not share semantic similarities assuming that the mapping function is smooth. In case the contrastive objective results in such as case, then we might have very noisy labeled data.

6. Section 3.1: "We train at batch sizes of either 1024 or 2048 using ADAM optimizer." -- These batch sizes are quite large than conventional (e.g. 32, 64). Is there a reason for that?

---

> ### Author Response · Authors · 2020-11-23
> **Response to Reviewer 1**
>
> We apologise to the Reviewer for being imprecise on several issues and hope that our approach is presented better in the revised version.
>
> Here our detailed response to the Main Comments:
>
> 1. We agree that almost all OOD approaches are based on feature extraction followed by binary classification, which is in fact the most natural approach to the problem. However, most other approaches assume or require that the in-distribution occupies a simply connected region in a lower dimensional (latent) space for optimal discrimination.  In contrast, our method does not require a separable latent space for OOD detection. In our approach, the output of the encoding function $f(x)$ can be a  ‘Swiss-cheese’ like latent space, where the in-distribution is mapped to 'holes' and the 'cheese' is OOD. The reason is that the mapping $h(x)=f(x)/||f(x)||$ projects in any case both in-distribution examples  and OOD examples onto the lower dimensional surface of a unit-hypersphere, where in-distribution examples and OOD examples can lie next to each other. As the discrimination is carried out relative to a reference example (nearest neighbour) and not by a fixed decision boundary, the distribution of OOD examples on the unit-hypersphere is not relevant.
>
> 2. We fully agree that we do not provide any evidence that our approach works outside the visual domain, although we can argue that contrastive methods have been successfully applied to NLP (Word2Vec) and Audio (Contrastive Predictive Coding). We therefore changed Title, Abstract, and the content of paper to narrow down our predictions to the visual domain.
>
> 3. We apologise for the ‘all’ in ‘… includes all semantic information’, which is certainly wrong and we have removed that. We can learn at most the information that is orthogonal to the transformations applied and although contrastive methods maximise the mutual information in theory, there is no sign that deep neural nets can approach this limit. However, we cannot follow the argumentation why our definition of ‘semantic neighbourhood’ is ill-defined. It is  a direct consequence of the transformations used to train $h(x)$ and the cardinality of the semantic neighbourhood (if it’s 4 or 32) has only a minor effect (Table 2 in revised version) but should be a small number to maximise the amount semantic information that can be used for discrimination. The transformations are well defined, at least for images of objects. The strength of transformations are chosen such that positive pairs $(x,x’)\sim P_{pos}$ get a higher score than any pair from the training set and negative pairs $(x,x’)\sim P_{neg}$ get a lower score than any semantic similar pair form the training set (see Fig.4 in the revised version).
>
> 4. Indeed the subnetworks likely share weights and are not independent. However, for an ensemble method to work that is not necessary. For example dropout as regularisation technique is effectively an ensemble method, averaging over exponentially many subnetworks during training that share weights. As the ensemble effect can indeed be expected to be larger for larger network size we use ResNet18/34 nets in the revised version and show the effect of our ensemble method in Appendix C.
>
> 5. That two examples not sharing much semantic information are found next to each other on the unit-hypersphere is indeed counter intuitive. The reason is that the neural network $f(x)$ puts out a $d$ dimensional vector, where OOD examples and training examples can be found in different regions, as intuitively expected. However,  $f(x)/||f(x)||$ projects $f(x)$ onto the lower dimensional surface of a unit-hypersphere with the effect that OOD and training examples can be mapped arbitrary close to each other, as the contrastive objective distributes examples almost uniformly across the unit sphere.
>
> 6. Contrastive objectives require large batch sizes to work well. For the discriminator we reduced batch size to 128.

---

> > ### Comment · AnonReviewer1 · 2020-11-24
> > **Paper has been made somewhat better than earlier**
> >
> > Some of my concerns have been addressed such as narrowing the scope as per the title, clarification on the degree to which the 'semantic information' has been captured by the model, and why close instances on the unit hyper-sphere might not share semantic information. I have increased my score by one point accordingly.
> >
> > However, I still find the number of datasets too few. I would encourage the authors to add additional datasets from at least one other domain (text/audio). It is easy to say (as mentioned in revised abstract) that the proposed technique can be extended widely to other types of data; in reality, it might be just very hard to define semantic neighborhood in an implementable manner for other types of data.

---

> > > ### Author Response · Authors · 2020-11-24
> > > **Response to Reviewer 1**
> > >
> > > As indeed we have not shown any results outside the visual domain in our paper, we will remove the assertion in the abstract that the approach "can be extended to a wide range of anomaly detection problems". However, we want to emphasise that our results within the visual domain show strong improvements for difficult OOD detection tasks.

---

### Official Review · AnonReviewer4 · 2020-10-29
**Recommendation to Accept**

**Rating:** 7
**Confidence:** 3

**Review:**

##########################################################################

Summary:

The authors present a new Algorithm for performing unsupervised anomaly detection in diverse applications such as visual, audio and text data. They propose a two-step method in which first they utilise contrastive learning in order to find a semantically dense map of the data onto the unit-hypersphere. Then, they classify neighbouring pairs of test examples as in- or out-of- distribution based on the amount of the shared semantic information. Finally, they show that in several anomaly detection problems in the field of visual data their proposed method outperforms several existing methods.

##########################################################################

Reasons for score:

I recommend to accept the paper since the authors deal with an important problem and they propose a clear and well-written method that outperforms in their empirical applications, at least, several existing approaches. Please find below cons that I suggest the authors to address in the rebuttal period.

##########################################################################

Cons:

1) Although the authors refer to several existing anomaly detection methods I would suggest to add a separate and relatively small literature review section in the paper. In that section the authors should list the most relevant, existing, anomaly detection methods and briefly explain them. This will improve the readability of the paper.

2) The authors identify that the main limitation of the proposed approach is the definition of a semantic similarity which in some applications can be very difficult. Therefore, I suggest the authors to perform a sensitivity analysis of their results with respect to the transformations that they use. I propose to add one or two tables similar to Table 1 in which they will compare versions of their method resulting from using different/misspecified transformations with the competing methods. They could for example add some 'noise' in the transformation that they use and re-perform the comparisons.

3) The authors should make, within their main text, reference to the Figures and the Algorithms that they present. By giving briefly the utility of each of their Figures and Algorithms they will improve substantially the readability of the paper.

##########################################################################

Minor comments:

1) Define d in 'd-dimensional' in page 2.

2) Conduct an extensive search for typos, correct for example the punctuation in 'everything that is not noise' at the bottom of page 7.

---

> ### Author Response · Authors · 2020-11-24
> **Response to Reviewer 4**
>
> We thank the Reviewer for the positive evaluation of our manuscript and for the helpful comments.
>
> Here our detailed response to the Major Comments:
>
> 1. We now included a literature review at the beginning of the paper and explain the methods that are most related to ours in more detail.
> 2. We now included the results of a sensitivity analysis (Table 2), where we changed transformation strengths and other hyperparameters of the model and reported the effect on AUROC values.
> 3. We now reference more consistently to Figs/Tables/Appendix to improve the readability of the paper.
>
> Minor Comments:
> 1.  we defined $d$ above Eq. 1
> 2.  we corrected the quotation marks

---

### Official Review · AnonReviewer2 · 2020-10-29
**Review AnonReviewer2**

**Rating:** 5
**Confidence:** 4

**Review:**


**UPDATE**

I acknowledge that I have read the author responses as well as the other reviews. I appreciate the clarifications and improvements made to the paper and have increased my score 5.

My concerns about the generality of the framework (as also pointed out by Rev1) still hold, however, as an evaluation on non-image data is still missing. I encourage the authors to extend their work further into this direction, but as is, I would keep my recommendation to reject.

#####

**Summary**

This work presents a generic approach for out-of-distribution (OOD) detection or anomaly detection (AD) called GenAD. GenAD consists of two steps: First, (i) learning a spherical representation via contrastive learning to capture semantic similarities, followed by (ii) training a classifier to discern between semantically similar and dissimilar pairs of samples, given the representation from (i). An experimental evaluation on in-distribution vs. out-of-distribution dataset pairs (CIFAR-10 vs. SVHN, CIFAR-10 vs. CIFAR-100, CIFAR-100 vs. CIFAR-10) is presented which shows that GenAD outperforms previous OOD methods on these settings.


**Pros**
+ OOD detection is an important open problem that is relevant and of interest to the community.
+ GenAD seems to improve over previous methods in the visual domain.
+ GenAD, in principle, is applicable to general types of data (e.g., images, audio, text, etc.).


**Cons**
- There are some critical details missing about the specific choices made for sampling negative pairs, which makes it hard to assess the technical correctness and merit of the presented approach. In general, I find it hard to follow and exactly understand all the relevant details from reading the description of the method in Section 2.
- Though the applicability of the approach to general types of data is emphasized, the experimental evaluation only includes image data.
- Some recent related work from the out-of-distribution [6, 9, 10, 5] and deep anomaly detection [7, 3, 1, 2, 4, 8] lines of research are missing which also study representations that are effective for detecting semantic out-of-distribution samples and propose various solutions.


**Recommendation**

As is, I recommend to reject this paper primarily due to a lack of clarity and missing details in the description of the approach, which makes it hard to assess the technical correctness and merit of GenAD.

In particular, how are $P_{pos}$ (via transformation or neighborhood or both?) and $P_{neg}$ exactly modeled in the experiments?
In Section 2.2, $P_{neg}$ is defined as the product of positive marginals, but how is this implemented?
How are the negative minibatch $\{x_k^r\}_{k=1}^N$ and the negative set of transformations in $T^{negative}$ in Algorithm 1 defined and chosen?

These details should be clarified and explained.


**Additional feedback and ideas for improvement**
- Include the missing details and try to explain the approach more clearly (there is one page of space currently left).
- Include other types of data in the experimental evaluation, which would strengthen the generality claim of the proposed approach.


**Minor Comments**

1. The title of the paper is very generic.
2. The figures in the paper are disproportionately large and waste quite some whitespace.
3. The batch sizes reported in the experiments are uncommonly large (1024, 2048). What is the reason for this choice?
4. I think Algorithm 2 can be removed, as it just describes $k$-NN using cosine similarity, right?
5. Section 1: ‘Note that it is possible for a datapoint to have high likelihood under a distribution yet be nearly impossible to be sampled, a property known as asymptotic equipartition property in information theory.’ Citation?
6. Section 1: ‘Intuitively, the OOD detection problem should be independent of the hardness of an in- distribution classification task.’ Why? I could imagine the hardness of an in-distribution classification task can be due to a complex in-distribution, for which the OOD detection problem is also more difficult.
7. Make use of page 8 in the main paper, e.g. move interesting claims and derivations to the main paper.
8. Section 3.1: ‘[...] - for both the encoder $f(x)$ and the *classifier* $s(x,x′)$.’ I would avoid to use the discriminator term.
9. Table 1: Add space between method names and citations.
10. Section 4: ‘[...], with increase in state-of-the-art AUROC from 0.783 to > 0.999.’ What about the 0.856 of OpenHybrid in Table 1?
11. Section 4: ‘Note that $h(x)$ *encodes features* of semantic similarity but not necessarily *features that allow* to score semantic dissimilarity.’
12. Section 4: ‘In fact, we observe for CIFAR-100 that examples from the same semantic neighbourhood do not always share the same label.’ Could you include some example images?


#####

**References**

[1] F. Ahmed and A. Courville. Detecting semantic anomalies. In AAAI, pages 3154–3162, 2020.

[2] L. Bergman and Y. Hoshen. Classification-based anomaly detection for general data. In ICLR, 2020.

[3] I. Golan and R. El-Yaniv. Deep anomaly detection using geometric transformations. In NeurIPS, pages 9758–9769, 2018.

[4] S. Goyal, A. Raghunathan, M. Jain, H. V. Simhadri, and P. Jain. DROCC: Deep robust one-class classification. In ICML, pages 11335–11345, 2020.

[5] P. Kirichenko, P. Izmailov, and A. G. Wilson. Why normalizing flows fail to detect out-of-distribution data. arXiv preprint arXiv:2006.08545, 2020.

[6] A. Meinke and M. Hein. Towards neural networks that provably know when they don’t know. In ICLR, 2020.

[7] L. Ruff, R. A. Vandermeulen, N. Görnitz, L. Deecke, S. A. Siddiqui, A. Binder, E. Müller, and M. Kloft. Deep one-class classification. In ICML, pages 4393–4402, 2018.

[8] L. Ruff, J. R. Kauffmann, R. A. Vandermeulen, G. Montavon, W. Samek, M. Kloft, T. G. Dietterich, and K.-R. Müller. A unifying review of deep and shallow anomaly detection. arXiv preprint arXiv:2009.11732, 2020.

[9] R. T. Schirrmeister, Y. Zhou, T. Ball, and D. Zhang. Understanding anomaly detection with deep invertible networks through hierarchies of distributions and features. arXiv preprint arXiv:2006.10848, 2020.

[10] Z. Wang, B. Dai, D. Wipf, and J. Zhu. Further analysis of outlier detection with deep generative models. arXiv preprint arXiv:2010.13064, 2020.

---

> ### Author Response · Authors · 2020-11-23
> **Response to Reviewer 2**
>
> We thank the reviewer for the fair and very helpful comments and for providing a list of references. Here our detailed response:
>
>  "As is, I recommend to reject this paper primarily due to a lack of clarity and missing details in the description of the approach, which makes it hard to assess the technical correctness and merit of GenAD."
>
> We apologize for the lack of clarity. We now provide the details which transformations are used for training $h(x)$ and $s(x,x’)$ and in particular how $P_{pos}/P_{neg}$ is defined in the main text and Appendix D.
>
> Response to Minor Comments:
> 1) We changed title and abstract and make clear that we apply our method to the visual domain.
> 2) Yes. We resized the distribution plot.
> 3) Large batch sizes are needed for contrastive learning (see Chen et al 2020), but indeed not necessary for the discriminator, which we changed to 128.
> 4) Yes.
> 5) We cited now Cover & Thomas, Elements of Information Theory.
> 6) Probably we have given a too strong statement here and therefore removed it. However, consider as in-distribution colored MNIST, with the constraint that only one color channel is allowed to be non-zero. Grey-scaled MNIST images are OOD by construction but label information doesn’t help detecting them if labels are independent of color.
> 8) If the Reviewer agrees, we would like to keep the discriminator terminology.
> 9) We correct the spacing
> 10) We compared with methods that don’t use labels but there was still an error, which we corrected.
> 11) We corrected that
> 12) We now added a new figures to show semantic similar images and transformation strengths.

---

### Official Review · AnonReviewer3 · 2020-11-02
**interesting work but needs more clarification/verification on methods/details to validate the results**

**Rating:** 5
**Confidence:** 4

**Review:**

Summary
- Presents GenAD as a general framework for anomaly detection
- Method builds on top of contrastive training and proposes to learn a discriminator to distinguish between semantically similar and dissimilar pair of examples
- Results are SOTA but need verification through code and methods clarification

Clarity/Quality:

Paper is overall written OK but several typos/grammatical errors as highlighted below:

- “For visual data we show new state-of-the OOD classification accuracies for standard benchmark data sets” -> new state of the
  “art” OOD classification
- “The contrastive objective aligns feature vectors h = h(x)” -> consider using different symbols for the vector output and the
   encoder function
- “A statically meaningful score” -> statistically?
- The notation “Pneg(x, x’) = Ppos(x)Ppos(x’)” is unclear. Is marginalization implied? (end of page 3)
- “mainly affects the weights of a small subnetwork Frankle & Carbin (2019)” - missing parentheses around reference
- “If we belief in the lottery hypothesis” -> belief to believe
- “We expect to see a significant increase in OOD detection performance upon increasing network size, which left to future work.”
 -> which is left to future work

Novelty:

Central claim - Contrastive training maps example to unit hypersphere but it is possible OOD examples can be in same neighborhood. Hence need a semantic discriminator and introduces it along with algorithms for sampling positives/negatives.

Significance:

The central idea is simple and well motivated. If (and it is a big if) the results are verified, this could be a very important paper in the field of OOD detection.

Questions/Comments/Clarification

- Assumption is all OOD is semantic which may not always hold true especially if there are stylistic varaiations introduced using different imaging equipment
- Unclear why gamma (Yneg and Ypos) was introduced
- Unclear how encoder and discriminator are trained? Is it jointly or separately? Are these the same networks? Architecture diagram for network setup is needed to clarify details
- Why does the discriminator enable learning of semantic dissimilarity?
- While algorithm for sampling of positives is specified how are negatives sampled?
- In Table 2, ablation corresponding to T(x) should be similar to results from (Winkens et al. 2020) right? However the
  corresponding values are much higher (78.3 vs 89.3). The only difference seems network sizes. Not sure how these results came
  about?
- In Appendix C2 - “To train the discriminator s(x, x0 ), we use almost the same network structure as our contrastive encoder but
  with smaller width and the MLP layer projects to a scalar output.” -> so do you not train the discriminator on top of contrastive
  representations? If yes, then how is the network pruned to smaller width?
- Why was ADAM used instead of LARS as in Chen et al?
- Claim of general framework for OOD detection is strong as no results shown on non visual domains.

Overall, this is an interesting idea but the method needs a lot more clarification and results need verification. Would encourage authors to share code to help verify the methods/results.

---

> ### Author Response · Authors · 2020-11-23
> **Reply to Reviewer 3**
>
> We thank the reviewer for the fair and helpful comments and spotting many tiny errors. Here our detailed response:
>
> 1) … idea is simple and well motivated. If (and it is a big if) the results are verified, this could be a very important paper in the field of OOD detection.
>
> In addition to providing the code for the paper, we now report in Table 1 AUROC values that are averaged over 2x5 runs for 2 standard deep neural network architectures (ResNet18 and ResNet34, 5 runs each). We further carried out a sensitivity analysis by changing different hyperparameters and types of transformations (Table 2). Our results are robust if transformations for P_pos(P_neg) are sufficiently moderate(strong), respectively.
>
> 2) Assumption is all OOD is semantic which may not always hold true …
>
> We agree that our method needs prior assumptions to design the type and strength of transformations used to train the model. However, the transformations define how the model generalizes over the in-distribution and therefore is part of the inductive bias that underlies any OOD model (e.g. Generative Models). The systematic tuning of hyperparameters using an in-distribution validation set can also be applied to our method by choosing transformation that maximise the ‘semantic’ similarity of nearest-neighbour pairs without rejecting an in-distribution test. But indeed, the resulting factors of variations might not be related to what is typically described as ‘semantics’.
>
> 3) Unclear why gamma (Yneg and Ypos) was introduced
>
> The effect of using gamma as a stochastic variable is now shown in Appendix C
>
> 4) Unclear how encoder and discriminator are trained
>
> We now wrote how training is carried out and what networks have been used in the Training Section and in Appendix D. In short, h(x) is readily trained before s(x,x’) gets the semantically close pairs determined by h(x).
>
> 5) Why does the discriminator enable learning of semantic dissimilarity?
>
> The examples that make up a negative pair for training the discriminator are transformations of two different examples from the training set – and the assumption is that transformations of two different examples are almost always more dissimilar than two independent transformations of the same example (if transformations are not extreme). See separation of the blue and red peak in Fig. 4.
>
> 6) While algorithm for sampling of positives is specified how are negatives sampled?
>
> We now clarify this point below Eq. 2.
>
> 7) In Table 2, ablation corresponding to T(x) should be similar to results from (Winkens et al. 2020) right?
>
> Winkens et al. 2020 use a different approach, so the values cannot be compared.
>
> 8) … so do you not train the discriminator on top of contrastive representations?
>
> We use two different Networks for training h(x) and s(x,x’) -- they use the same trunk architecture (ResNet18/34) but don’t share parameters. We clarify this now in the 'Training' section.
>
> 9) Why was ADAM used instead of LARS as in Chen et al?
>
> We tried LARS but don’t saw any improvement. So, we stayed with ADAM (or AMSGrad to be precise)
>
> 10) Claim of general framework for OOD detection is strong as no results shown on non visual domains
>
> We agree. We changed title and Abstract to make this clear from the beginning.

---

### Author Response · Authors · 2020-11-24
**Change in title**

As there was common agreement among the reviewers that the generality promised by the title is not supported by the results in the manuscript we decided to change the title and consequently the name of our method.

---

### Decision · Program_Chairs · 2021-01-07
**Final Decision**

**Decision:**

Reject

**Comment:**

The paper proposes a two stage approach for anomaly detection - first train a low dimensional embedding potentially using self-supervised learning methods, and then train a discriminator on top of the embedding that takes in pairs of examples and outputs a score which can be used for anomaly detection. A test example is paired with the next nearest neighbor. A common concern of the reviewers was on the claim of the paper to be a general approach for anomaly detection whereas experiments are reported only on vision datatsets. The authors have addressed this by making changes to the title and to the claims made in the paper. However R1 and R2 still have concerns about insufficient empirical evaluations, in particular lack of non-vision datasets.

As the paper aims to tackle the problem where OOD examples are spread through the sphere, appearing mixed with normal examples, I think fitting a nonparametric density model (eg, using KDE) or parametric density model (eg, a mixture model) on the embeddings is a natural baseline to compare with.

I encourage the authors to strengthen the empirical section of the paper based on reviewers' comments and resubmit to a future venue.